# Substantial Na-Ion Storage at High Current Rates: Redox-Pseudocapacitance through Sodium Oxide Formation

**DOI:** 10.3390/nano12234264

**Published:** 2022-11-30

**Authors:** Engelbert Portenkirchner

**Affiliations:** Institute of Physical Chemistry, University of Innsbruck, 6020 Innsbruck, Austria; engelbert.portenkirchner@uibk.ac.at; Tel.: +43-512-507-58014

**Keywords:** batteries, Na-ions, pseudo-capacitive, surface films, oxygen diffusion, sodium oxide, titanium dioxide

## Abstract

Batteries and supercapacitors, both governed by electrochemical processes, operate by different electrochemical mechanisms which determine their characteristic energy and power densities. Battery materials store large amounts of energy by ion intercalation. Electrical double-layer capacitors store charge through surface-controlled ion adsorption which leads to high power and rapid charging, but much smaller amounts of energy stored. Pseudocapacitive materials offer the promise to combine these properties by storing charge through surface-controlled, battery-like redox reactions but at high rates approaching those of electrochemical double-layer capacitors. This work compares the pseudo-capacitive charge storage characteristics of self-organized titanium dioxide (TiO_2−x_) nanotubes (NTs) to flat TiO_2−x_ surface films to further elucidate the proposed charge storage mechanism within the formed surface films. By comparing TiO_2−x_ NTs to flat TiO_2−x_ surface films, having distinctively different oxide mass and surface area ratios, it is shown that NaO_2_ and Na_2_O_2_ formation, which constitutes the active surface film material, is governed by the metal oxide bulk. Our results corroborate that oxygen diffusion from the lattice oxide is key to NaO_2_ and Na_2_O_2_ formation.

## 1. Introduction

Empirically grounded technology forecasts call for a rapid transition towards a decarbonized, global energy supply. Scenarios are outlined which may become critical for addressing climate change while at the same time will likely result in trillions of dollars net energy cost savings [1]. Storage of intermittent, renewable energy is thereby considered a key technology for realizing the anticipated energy transition [2,3,4]. Particularly, rechargeable batteries have emerged as an ideal storage technology for electrical energy. In the last decades, rechargeable batteries became an indispensable part of numerous applications, that range from small-scale electronic devices to high-power electric vehicles [5,6]. The lithium (Li)-ion battery, being the most deployed rechargeable battery technology on the market, has raised concerns, mainly due to its environmentally harmful mining and its limited availability [7]. These issues resulted in a substantial increase in the price of Li which is expected to further rise as the transition towards electro-mobility will pose substantial stress upon the resource sector [8]. Consequently, alternative battery technologies based on sodium (Na) over Li have been the focus of research for many years, inspired by the high natural abundance of Na [8,9,10].

State-of-the-art battery materials store large amounts of energy (~300 Wh kg^−1^) throughout the bulk of the active material by ion intercalation, which is a diffusion-limited, faradaic reaction. Diffusion-limited redox reactions are often slow and, although these redox reactions lead to high energy density, charging is slow and requires several minutes to hours [11]. Electrical double-layer capacitors, on the other hand, are not governed by faradaic charge storage. Differently, these devices store charge through surface-controlled ion adsorption. Such a capacitive charge-storage mechanism enables high power and consequently allows for rapid charging which is in the order of minutes, but double-layer capacitors store much smaller amounts of energy [12]. Recently, a new class of materials has received great interest, commonly known as pseudocapacitive materials. These materials undergo surface-controlled, faradaic reactions, which, because they are confined to the surface, demonstrate charging rates comparable to those of double-layer formation. Over the past years, several materials were found that exhibit pseudocapacitive charge storage, mainly metal oxides (e.g., RuO_2_ and MnO_2_) [13], metal sulfides (e.g., MoS_2_) [14], metal hydroxides (e.g., NiOH) [15], conducting polymers (e.g., polyaniline) [16,17], and redox-active organic molecules (especially quinones) [18,19,20].

Our group was able to show that electrochemically grown, oxygen-deficient, carburized, and self-organized titanium dioxide (TiO_2−x_) nanotubes (NTs) are capable of substantial Na-ion storage in the range of 200 mAh g^−1^ at a current rate of 30 mA g^−1^ (C/20) or 60 mAh g^−1^ at high current rates of 20 C (12 A g^−1^) [21]. Furthermore, different to the material performance in a Li-ion-containing electrolyte [22,23], the Na-ion storage capacity self-improves considerably as cycling proceeds [21]. It was found that sodium superoxide (NaO_2_) and sodium peroxide (Na_2_O_2_), formed at the electrode surface in an acicular surface film, are the main constituents of the charge-storage products. The Na-ion storage is thereby dominated by pseudocapacitive charge storage at high sodiation rates [24]. Furthermore, in a very recent publication, it has been demonstrated that this surface chemistry is not unique to TiO_2_ NTs, but can be seen as a more general characteristic for Na-ion storage at metal oxide surfaces. Since the initial discharge product in sodium–oxygen batteries is also NaO_2_, which undergoes dissolution and subsequent transformation to Na_2_O_2_ and Na_2_O_2_ dihydrate, these findings became particularly interesting in the field [25,26,27].

The prevailing Na-ion storage mechanism has been discussed in great detail in a previous publication, where it was assumed that partly mobile oxygen atoms in the metal oxide lattice are the main oxygen source for NaO_2_ and Na_2_O_2_ formation. Whether the necessary oxygen for the charge storage truly originates from the bulk oxide or from an undesired decomposition of the electrolyte has since then been a major issue of debate. In this work, by comparing the Na-ion storage characteristics of TiO_2−x_ NTs to flat TiO_2−x_ surface films, having distinctively different oxide mass and surface area ratios, it is shown that the NaO_2_ and Na_2_O_2_ formation is governed by the metal oxide bulk. This is further corroborating the idea that the oxygen source mainly originates via oxygen diffusion from the lattice bulk material.

## 2. Materials and Methods

Synthesis: Synthesis of the TiO_2−x_ flat surface films and TiO_2−x_ NTs have been performed by a slightly modified procedure previously reported in reference [22,28]. Both systems were grown electrochemically on a mechanically polished (SiC grinding papers P1200, P2500, P4000, Buehler, Carbimet) titanium (Ti) metal disk (18 mm Ø, 1 mm thickness, Advent, 99.6%) followed by cleaning in an ultrasonic bath (Bandolin, Sonorox, Berlin, Germany) in ethanol. Just before anodization, the disks were etched for 5 min in 0.1 M HNO_3_ (VWR International, AnalaR Normapur, 65%, Vienna, Austria). Electrochemical oxidation to an amorphous TiO_2_ film is performed in a two-electrode setup with a copper plate as a current collector, the titanium disc as a working electrode, a platinum net as the counter electrode, and 0.1 M H_2_SO_4_ as the electrolyte, connected to a DC power supply (EA-PSi 6150-01, 150 V/1.2 A). After cell assembly, a constant potential of 20 V was applied for 60 min with an initial voltage ramp of 1 V s^−1^. The TiO_2_ NTs are synthesized by electrochemical anodization in an electrolyte containing 50 vol% ethylen glycol (VWR, AnalaR Normapur, 99.7%) and 2 wt% NH_4_F (Alfa Aesar, NH_4_F∙0.5H_2_O, 99.9975%) in deionized water and also by applying a potential of 20 V for 60 min after an initial potential ramp of 1 V s^−1^ from 0 V to 20 V. Right after the anodization, the electrolyte was removed by rinsing with deionized water followed by drying in air. For converting the as-grown amorphous oxide and NTs to anatase TiO_2−x_ oxide and NTs, they were thermal annealed in Ar at 400 °C. The quartz tube reactor was purged with 600 sccm Ar (Messer, 5.0) at room temperature for 75 min in order to remove excess air. The Ar flow was then reduced to 200 sccm. The temperature has subsequently been gradually increased by 10 °C min^−1^ up to 200 °C, 5 °C min^−1^ up to 300 °C and 3 °C min^−1^ up to 400 °C. The temperature was kept constant at 400 °C for 340 min followed by a not-rate-controlled cooling phase towards room temperature (i.e., 22 °C), resulting in anatase oxygen-deficient TiO_2−x_. Structural and chemical characterization of the synthesized electrodes has been carried out by Raman (Appendix A), XRD (Appendix A), and XPS (Appendix A) measurements for both, flat and compact TiO_2−x_ surface film electrodes and electrodes with TiO_2−x_ nanotubes.

Electrochemistry: For the electrochemical characterization, a three-electrode EL-Cell setup (ECC-Ref Cell) was used. Therefore, a 0.1 to 0.5 mm thick Na metal foil (99.9%) counter electrode (CE) having a diameter of 16 mm and an Na reference electrode (RE) were prepared and mounted in the PEEK sleeve which in turn is placed into the stainless-steel cell base. The glass fiber separator (EL-Cell, 18 mm × 1.55 mm) is inserted into the PEEK sleeve and 500 µL of the 99.9% pure electrolyte, containing 1 M NaFSI (Bis(fluorosulfonyl)imide) in 1:1 V/V ethylene carbonate:dimethyl carbonate from Solvionic, is added [29]. The whole assembling process has been carried out in an Ar-filled glove box with H_2_O and O_2_ levels below 0.1 ppm. The galvanostatic sodiation/desodiation was carried out between 3.0 and 0.1 V vs. Na/Na^+^ until the current increase, described by a logistic growth function [24], levels off at 117, 59, 24, 12, 6 µA for the TiO_2−x_ flat surface films and 200 µA, 40 µA, 20 µA, and 10 µA for TiO_2−x_ NTs.

Secondary electron microscopy: SEM cross-sections were prepared using a Jeol IB-19530 CP Ar plasma etcher and subsequently, imaged by a Jeol JSM-7610F field emission SEM. The electrochemical oxidation of the TiO_2−x_ was realized with a 0.125 mm thick titanium foil instead of the 1 mm thick titanium disc. Then, the sample was put in the path of a 2 mm broad beam of stationary Ar plasma. A shielding plate protected the samples and only the part protruding from the edge of the mask is milled away. This results in a clean polished cross-section. The sample was cut from the backside in order to protect the surface of the thin film and to improve the cutting edge of the film.

## 3. Results

For Ti and other so-called valve metals, a compact oxide layer can be grown by anodization in an aqueous electrolyte, forming compact oxide layers up to several 100 nm thick [30]. In this work, titanium(IV)-oxide electrodes are easily synthesized by the electrochemical oxidation of the parent Ti-metal substrate. If fluoride ions are present in the electrolyte, anodization of a Ti metal substrate leads to the formation of an array of self-organized, well-aligned TiO_2_ NTs [31,32]. Thermal annealing at a temperature of 400 °C in an argon (Ar) atmosphere, results in the conversion of the as-grown amorphous oxide to oxygen-deficient anatase TiO_2−x_ (x < 2) [33]. Oxygen vacancies formed during this process are found to enhance the charge-transfer properties and ion diffusion of these electrodes and to assist a potential phase transition resulting from ion intercalation and de-intercalation. The effect of such reductive thermal annealing on the TiO_2_ NT array anodes for Li-ion batteries, has been previously investigated by our group [22,23,34]. This has been found to allow for higher Li-ion intercalation capacities and rate capabilities compared to stoichiometric anatase NTs [22,35,36,37].

While Li-ion intercalation in TiO_2−x_ is well known and studied, with a theoretical capacity of anatase or rutile TiO_2_ being 335 mAh g^−1^ [28,31,38], Na-ion intercalation in TiO_2−x_ is still under debate [21,39,40]. In relation, it has been shown that TiO_2−x_ bulk films and NTs are capable of substantial Na-ion storage, which self-improves as cycling proceeds [21]. Figure 1 shows these self-improving Na-ion storage characteristics for TiO_2−x_ NTs. The specific gravimetric capacity increases from initially 25 mAh g^−1^ to 84 mAh g^−1^ after 230 galvanostatic sodiation/desodiation cycles between 3.0 V and 0.1 V (Figure 1a).

Much to our surprise, it has been found in a previous work that this self-improved Na-ion storage is retained after the initial formation process, allowing unprecedented fast sodiation/desodiation rates of 20C (12 A g^−1^) for Na-ion storage using TiO_2−x_ NTs [21]. Such high-rate capabilities are considered impossible to be achieved through Na-ion intercalation and could not be observed for the Li-ion intercalation in TiO_2−x_ NTs, expected to be much faster compared to the Na-ion intercalation.

Related to this self-improving Na-ion storage characteristics, also a distinctive change in the cyclic voltammetry (CV) measurements is observed before (Figure 1b) and after (Figure 1c) film formation (i.e., after 230 galvanostatic sodiation/desodiation cycles) at a scan rate of 1 mV s^−1^. In the initial CV measurement, no distinct features that are related to Na-ion storage are visible, except a significant cathodic current increase at potentials below 0.5 V. After galvanostatic sodiation/desodiation cycling, however, broad anodic and cathodic peaks are observed in the TiO_2−x_ NTs electrodes, with anodic and cathodic peak maxima located at 0.85 V and 0.73 V, respectively. In accordance with the exceptionally high sodiation/desodiation rates in the GCPL measurements, these new redox features are contained at various scan rates measured from 0.05 to 200 mV s^−1^ [21].

The current that is measured during CV comprises the sum of stored charge that originates from both, faradaic and non-faradaic processes.

The non-faradaic process arises mainly from contributions related to the formation of the electrochemical double layer. The faradaic contributions, on the other hand, originate from Na-ion insertion into the TiO_2−x_ NTs host matrix, together with charge transfer processes that are limited to reactions confined to the surface, also known as pseudocapacitance [41]. Further details of the CV scan rate dependence allow us to quantitatively extract the capacitive contribution of the current response. This method can be used to describe the current response at a certain potential as the combination of surface (pseudo-capacitive) and insertion (bulk) processes [42,43]. A more detailed description of this method is reported in our previous publication [21]. The capacitive contribution, at a scan rate of 1 mV s^−1^, is shown in Figure 1e, indicating that the majority of the charge measured (76%) is capacitive in nature. The remaining sodiation charge is mainly created at the peak potentials, indicating that the potential Na-ion insertion takes place in the potential range of the peaks [39,44]. The analysis at different scan rates shows that the capacitive contribution progressively increases with increasing scan rate (Figure 1d) up to 94% capacitive current at a scan rate of 20 mV s^−1^. The Na-ion charge storage is therefore governed by pseudo-capacitive characteristics, allowing for the excellent rate capabilities and storage capacities measured [21,24]. The prevailing Na-ion storage mechanism, as elucidated in a previous publication, shows that mainly inorganic compounds, such as NaO_2_, Na_2_O_2_, and NaCO_3_, are the main constituents formed at the electrodes surface with a characteristic acicular morphology (Figure 2) [24].

It has been assumed that partly mobile oxygen atoms in the metal oxide lattice are the main oxygen source for NaO_2_ and Na_2_O_2_ formation. Oxygen is, at first, the limiting factor and its diffusion through the anatase, oxygen-deficient, TiO_2−x_ crystal structure, as indicated in Figure 2b, is expected to be a very slow process. Since oxygen is strongly bound in TiO_2_ and an initial oxygen vacancy formation is necessary for oxygen diffusion [45], the initial source of the oxygen is crucial for the surface film formation process. Whether the necessary oxygen for the charge storage truly originates from the bulk oxide or from an undesired decomposition of the electrolyte has since then been an issue of debate [26,46,47].

If the oxygen that is limiting the charge storage originates from an external source—i.e., electrolyte decomposition, side reactions with the Na metal counter electrode and/or impurities (leakages) of the battery cell—then the active surface area of the electrode should be the critical parameter governing the charge storage process. If, on the other hand, oxygen diffusion from the metal oxide lattice is prevailing, characterized by a slow (solid-solution type) oxygen diffusion, the amount of bulk oxide material should determine the observed current response. Following this line of reasoning, the Na-ion storage characteristics of TiO_2−x_ NTs in comparison to flat TiO_2−x_ surface films, having distinctively different oxide mass and surface area ratios (Figure 3), is investigated.

Figure 3 shows the schematic process for electrode preparation by electrochemical oxidation of the parent Ti-metal substrate, a metallic titanium disk with a diameter of 18 mm and a thickness of 1 mm. By anodization in an aqueous electrolyte, applying a constant voltage of 20 V for 60 min, a compact oxide layer is grown with approx. 200 nm thickness (Figure 3c,e). Differently, if fluoride ions are present in the electrolyte, anodization forms highly ordered, self-organized TiO_2_ NTs (Figure 3d) [48]. The processes involved have been intensely studied and mostly understood regarding the prevailing mechanism [49,50]. From cross-section SEM micrographs (Figure 3e), 1.1 µm average tube length, 115 nm average pore diameter and a solid hemisphere at the bottom of each tube are determined. With this approach, two distinctively different morphologies are formed for the thick and compact solid TiO_2−x_ layer electrodes and the self-organized TiO_2−x_ NTs. These two electrode geometries differ by a ratio of about 80 (exactly 82.6) with respect to their surface area and by a ratio of about 3 (exactly 2.9) with respect to their TiO_2_ oxide mass (Figure 4c,d). Since galvanostatic long-term cycling has been identified to be essential for initiating the self-improving in the charge storage process, both types of electrodes, TiO_2−x_ layer electrodes and the self-organized TiO_2−x_ NTs, were subject to galvanostatic cycling with potential limitation (GCPL, i.e., Figure 1a) performed in the voltage range between 0.1 to 3 V at different specific current densities, starting with approx. 50 µA cm^−2^ down to 0.5 µA cm^−2^ related to the electrode’s surface area.

Figure 4 shows the electrochemical characterization and comparison by CV measurements for the initial (dashed line) and aged (solid line, after film formation) compact TiO_2−x_ layer and self-organized TiO_2−x_ NT electrodes. Corresponding electrochemical impedance spectroscopy (EIS) measurements of TiO_2−x_ NTs before (initial) and after (aged) film formation are given in the Appendix A. As seen previously for TiO_2−x_ NTs (Figure 1c) these self-improving Na-ion storage characteristic distinctively change the current response in the CV measurements. Remarkably though, for both electrode types of TiO_2−x_ layer electrodes (Figure 4a) and the self-organized TiO_2−x_ NTs (Figure 4b), the change in CV response observed before and after film formation (that is after 230 galvanostatic sodiation/desodiation cycles) is very similar in form and shape. Both electrode types show, after self-improvement, the characteristic, broad peak pair with their peak maxima located at about 0.85 V and 0.70 V, respectively. What is distinctively different though is their peak current maxima, being 62 μA for the flat TiO_2−x_ layer electrodes (Figure 4a) and 192 μA for the self-organized TiO_2−x_ NTs (Figure 4b). The measured ratio of the peak current maxima, being 3.1 clearly reflects the ratio in TiO_2_ oxide mass being 2.9, and not the ratio in the surface area being 84.1. The CV measurements for both electrode types at a faster scan rate of 20 mV s^−1^ before and after film formation are very similar in form and shape compared to the one at 1 mV s^−1^ (Appendix A). Both electrodes show again the broad peak pair with their peak maxima slightly shifted to about 1.08 V and 0.68 V for the TiO_2−x_ layer electrodes and to about 1.05 V and 0.64 V for the TiO_2−x_ NTs, respectively.

This clearly shows that the bulk oxide determines the self-improving charge storage characteristics, suggesting a solid-solution-type oxygen diffusion through the metal oxide in order to form the NaO_2_ and Na_2_O_2_-containing, acicular surface film. Since oxygen diffusion through the bulk metal oxide is known to be kinetically slow, nanostructuring, i.e., by nanotube formation, definitely enhances the surface film formation kinetics by lowering the diffusion pathways for activated oxygen to the electrode’s surface. In addition, for porous nanostructured electrodes (like TiO_2_ NTs), next to a good electronic conductivity [37], a low pore resistance is considered important in order to achieve high-charge transfer kinetics, necessary for pseudocapacitive Na-ion charge storage [51].

## 4. Conclusions

In this work, two different electrode geometries, titanium dioxide flat surface films (TiO_2−x_) and self-organized titanium dioxide nanotubes (TiO_2−x_ NTs) have been investigated and compared towards their Na-ion storage characteristics. Both systems showed the previously reported, self-improving effect for Na-ion storage capacity due to a surface film formation containing sodium superoxide (NaO_2_) and sodium peroxide (Na_2_O_2_) as the active storage materials. Different to conventional rechargeable batteries that are governed by slow intercalation chemistries, these surface films allow for fast sodiation and desodiation rates, due to their pseudo-capacitive redox reactions. As a key component for the initial film formation, mobile oxygen atoms were identified. The two different electrode geometries investigated are characterized by distinctively different surface area to oxide mass ratios. These differences in texture and mass allow for a clear correlation of the self-improved peak currents during CV measurements to the amount of metal oxide bulk material. Hence, the presented results fully corroborate the previously proposed film formation mechanism claiming that the oxygen source originates from oxygen diffusion of the lattice bulk oxide. Since the surface film formation chemistry, which is related to the self-improving charge storage, is not unique for TiO_2_, but most likely serves as a common scheme for Na-ion storage at metal oxide surfaces, these findings are regarded important to further advance Na-ion and Na-oxygen batteries in general.

## Figures and Tables

**Figure 1 nanomaterials-12-04264-f001:**
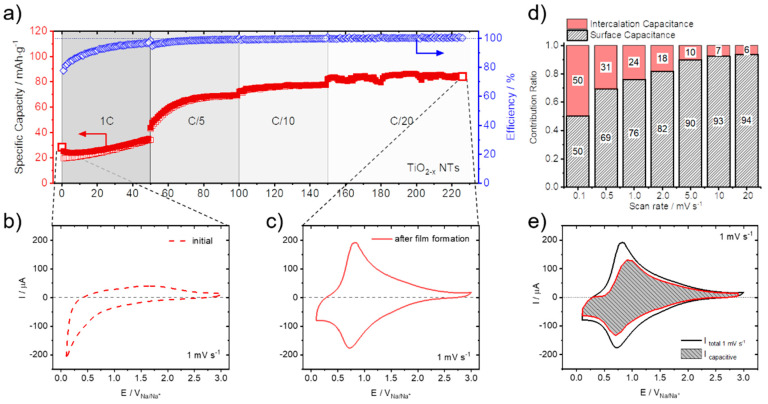
Self-Improving Na-ion storage: (**a**) Specific gravimetric capacities and Coulombic efficiencies versus sodiation/desodiation cycle number of TiO_2−x_ NTs, measured at different C-rates (from 1C to C/20) between 3.0 V and 0.1 V (sodiation: filled squares / desodiation empty squares). (**b**) CV measured before and (**c**) after film formation at a scan rate of 1 mV s^−1^. (**d**) Contribution ratio of capacitive and insertion charge versus scan rate. (**e**) Separation of capacitive (gray shaded area) and total (black line) current at a scan rate of 1 mV s^−1^. Measurements were performed in the Na-containing electrolyte (see Section 2). Reproduced and modified in part with permission from reference [21].

**Figure 2 nanomaterials-12-04264-f002:**
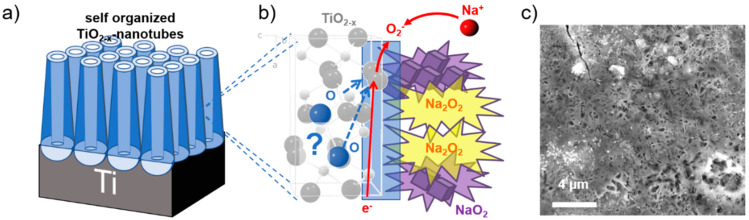
Surface film formation: (**a**) Scheme of the TiO_2−x_ NT array and (**b**) the corresponding anatase crystal structure with the proposed reaction of Na-ions and oxygen to sodium peroxide (Na_2_O_2_) and/or sodium superoxide (NaO_2_). (**c**) SEM top view image of TiO_2−x_ NTs after 230 galvanostatic cycles between 3.0 V and 0.1 V showing the surface coverage after sodiation by an acicular surface film. Reproduced and modified in part with permission from reference [24].

**Figure 3 nanomaterials-12-04264-f003:**
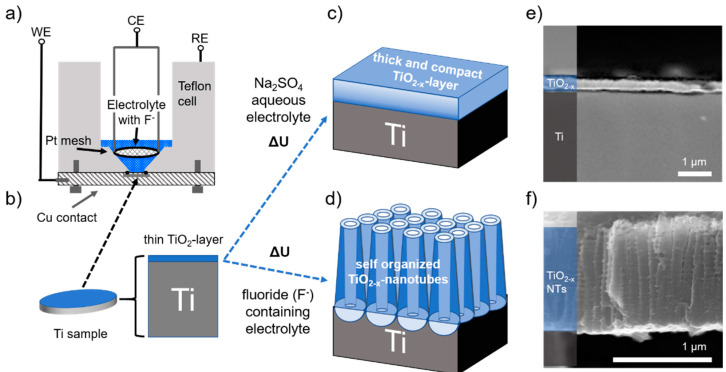
Scheme of the electrode formation and morphological characteristics. (**a**) Schematic of the electrochemical cell used for the anodization of a titanium metal disk shown in (**b**) as the working electrode. Anodization forms either a thick and compact solid TiO_2−x_ layer (**c**) on the metal surface, or, in a fluoride-containing electrolyte (**d**), self-organized TiO_2−x_ NTs. Cross-section SEM images of (**e**) the 200 nm thick oxide film and (**f**) approx. 1 μm long TiO_2−x_ NTs formed. The different layers are color-highlighted and labeled on the left side.

**Figure 4 nanomaterials-12-04264-f004:**
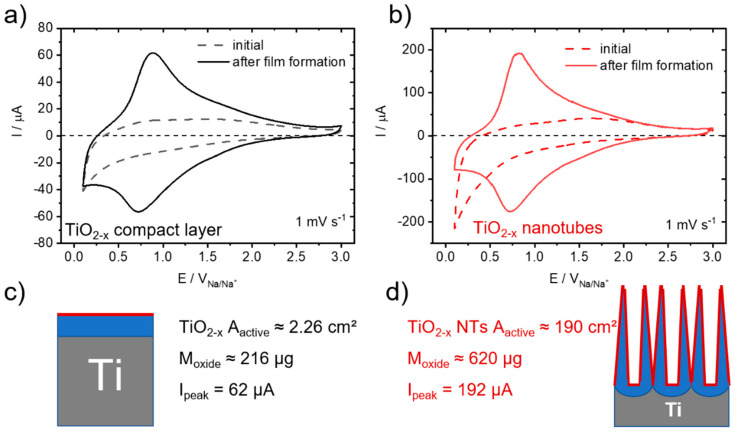
Electrochemical characterization and comparison. CV measurements for the initial (dashed line) and aged (solid line, after film formation) electrodes with (**a**) a compact TiO_2−x_ layer and (**b**) self-organized TiO_2−x_ NTs. Scheme and key parameters measured for (**c**) a compact TiO_2−x_ layer and (**d**) self-organized TiO_2−x_ NTs.

## Data Availability

The data presented in this study are available on request from the corresponding author.

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
