# Peer review of "Substantial Na-Ion Storage at High Current Rates: Redox-Pseudocapacitance through Sodium Oxide Formation"

_nanomaterials, 2022, doi:10.3390/nano12234264_

Round 1

Reviewer 1 Report

The manuscript reported the fabrication of TiO2-x flat surface films and TiO2-x nanotubes(NTs), and  compared the pseudo-capacitive charge storage characteristics of TiO2-x NTs to flat TiO2-x surface films to elucidate the proposed charge storage mechanism within the formed surface films. It seems that this work has been partially published somewhere, hence the structure characterization info of the materials is missing. Additionally, the performance comparison such as specific capacity, cycling and rate performance between two materials with different morphology are not provided in this manuscript. As this work does not reveal significant novelty, it cannot be recommended for publication in Nanomaterials.

Reviewer 2 Report

Review comment.

This paper compares two types of TiO2-x interface towards the clarification on the rate determining step of the surface pseudocapative portions of the sodium ion storage. The paper is interesting and it can be accepted after addressing the following issues.

1.     I’m a bit surprised on the electronic conductivity of the NT type TiO2-x, even though there is a barrier layer between the tube and the remaining Ti substrate. What is the thickness of the barrier layer?

2.     How is the electrode weight determined in this work? Also, the size of the working electrode should be described.

3.     Why the Na+ cannot intercalate into the TiO2-x matrix in the initial cyclings? Does Na+ eventually intercalate into TiO2-x matrix in the later cycles? Or only the generation of NaO2, Na2O2, Na2CO3 gradually enhanced the capacity of the system? How does the shape of CV changed as the cell was cycled?

4.     How is the CV profile look like at 20 mV s-1 (for any of the two types of TiO2-x)?

Round 2

Reviewer 1 Report

The manuscript quality has been improved, so it can be accepted for publication in Nanomaterials.